# Test-time Augmentation for Factual Probing

**Go Kamoda**[1]   **Benjamin Heinzerling**[2,1]   **Keisuke Sakaguchi**[1,2]   **Kentaro Inui**[3,1,2]

[1]Tohoku University   [2]RIKEN   [3]MBZUAI

go.kamoda@dc.tohoku.ac.jp   benjamin.heinzerling@riken.jp
keisuke.sakaguchi@tohoku.ac.jp   kentaro.inui@mbzuai.ac.ae

## Abstract

Factual probing is a method that uses prompts to test if a language model "knows" certain world knowledge facts. A problem in factual probing is that small changes to the prompt can lead to large changes in model output. Previous work aimed to alleviate this problem by optimizing prompts via text mining or fine-tuning. However, such approaches are relation-specific and do not generalize to unseen relation types. Here, we propose to use test-time augmentation (TTA) as a relation-agnostic method for reducing sensitivity to prompt variations by automatically augmenting and ensembling prompts at test time. Experiments show improved model calibration, i.e., with TTA, model confidence better reflects prediction accuracy. Improvements in prediction accuracy are observed for some models, but for other models, TTA leads to degradation. Error analysis identifies the difficulty of producing high-quality prompt variations as the main challenge for TTA.

🜲 github.com/gokamoda/TTA4FactualProbing

## 1 Introduction

Pre-trained language models (LMs) such as BERT (Devlin et al., 2019) and T5 (Raffel et al., 2020) implicitly encode world knowledge from the training corpus in their parameters (Roberts et al., 2020). Encoded knowledge can be retrieved from an LM via a suitable prompt (Petroni et al., 2019). For example, a prompt such as "Where did Albert Einstein die?" is designed to retrieve the fact that "Albert Einstein died in Princeton." However, this capability is not robust since small changes to the prompt can lead to drastic output changes (Heinzerling and Inui, 2021; Cao et al., 2022). If the model fails to answer correctly, it is thus difficult to distinguish if it did not learn the corresponding fact during pre-training or if it actually did but did not produce the correct answer with the given prompt.

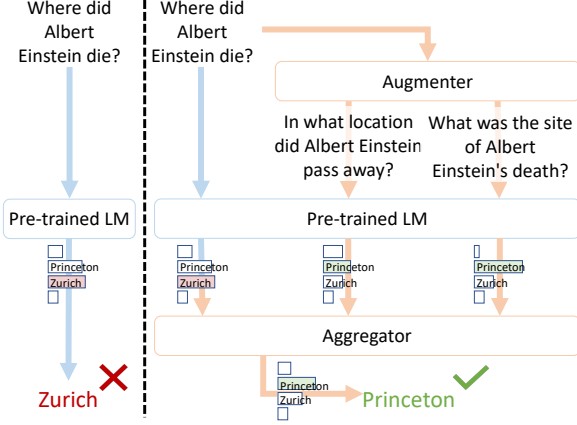

Figure 1: Factual probing with a single prompt (left) and with test-time augmentation (TTA). The orange components are added when using TTA. The Augmenter automatically augments the original prompt. The aggregator takes the generations from all prompts as input and outputs the generation with the highest score.

Prior work has aimed at finding better prompts for factual probing[1], typically by employing supervised learning to find an optimal input token sequence for a given relation (Shin et al., 2020; Jiang et al., 2020; Zhong et al., 2021). Since these approaches require supervision for each relation, they do not generalize to unseen relation types, which reduces practical applicability.

Here, we propose to use test time augmentation (TTA) as an unsupervised, relation-agnostic approach for improving prompt-based factual probing. TTA originates in computer vision, where given an input image at test time, the idea is to 1) apply augmentations such as flipping, cropping, or contrast changes, 2) have the model make predictions for all augmented versions of the image, and then 3) aggregate these predictions to obtain the final prediction. TTA has been found to increase robust-

---

[1]While "probing" more commonly refers to model analysis via light-weight classifiers, we follow prior work in using "factual probing" to denote model analysis via knowledge-eliciting prompts.

ness and to improve model calibration by reducing overconfidence in wrong predictions (Krizhevsky et al., 2012; Wang et al., 2019; Perez et al., 2018; Matsunaga et al., 2017). The benefits are also desirable in factual probing, where LMs should exhibit robustness to paraphrases and should generate well-calibrated predictions. However, TTA has found little use in NLP so far. While it is relatively easy to create feature-preserving augmentations of images (e.g., a flipped image of a cat is still an image of a cat), meaning-preserving augmentation of text is a challenging problem since even small edits, such as replacing a single token, can completely change the meaning of a sentence (Rethmeier and Augenstein, 2023). In this paper, we empirically demonstrate the benefits and challenges of TTA for factual probing.

As a similar approach in the context of chain-of-thought reasoning, Wang et al. (2023) prepare a single prompt as model input and aggregate multiple outputs ("reasoning paths") to improve model performance. TTA differs from this method in that it automatically prepares multiple inputs, i.e., prompts.

To apply TTA to factual probing, we add a prompt augmenter and a prediction aggregator to the prediction process (Figure 1). First, the input prompt is automatically augmented by the augmenter. The augmented prompts are then individually fed to an LM. The aggregator collects the outputs for each prompt and determines the final prediction. Our evaluation of this setup consists of two parts: We 1) evaluated the overall prediction accuracy and investigated the impact of the number of augmented prompts on the accuracy, and 2) inspected the change in the confidence of model predictions.

Results showed that the greater the number of augmented prompts, the better the performance when implementing TTA. TTA was also effective in reducing the number of overconfident and incorrect outputs. However, in terms of overall accuracy, TTA was only effective on smaller models.

## 2 Setup

**Dataset** We constructed a dataset of 12,500 relational facts from Wikidata. Each fact is composed of a subject, a relation, and an object. We filtered out facts with multiple possible objects and collected 500 unique facts for each of the 25 relations[2].

---

[2] The relations we selected are shown in the appendix.

For each relation, we manually created an English prompt template, e.g., "Where did {subject} die?"

**Augmenter** To yield paraphrases with variety, the augmenter uses three types of prompt augmentations. The first type is synonym replacement, which replaces words in the input prompt with a synonym. For instance, this type of augmentation replaced the word "buried" with "inhumed". Synonyms are selected via word embedding similarity using GLoVe (Pennington et al., 2014) or via Word-Net (Saedi et al., 2018). The second augmentation type is back-translation, with French, Russian, German, Spanish, and Japanese as target languages. The third augmentation type is stopword filtering, which removes stopwords from the prompts.

From a single original prompt, the augmenter produces one prompt via stopword filtering and four prompts for each of the seven other augmentation methods, resulting in a total of 30 prompts.

**Model** We ran experiments on the following pre-trained language models: T5 for Closed Book Question Answering (Small, Large, 3B, 11B) (Roberts et al., 2020), FLAN-T5 (Small, XL) (Wei et al., 2022), and T0_3B (Sanh et al., 2022).

Models decode with beam-search where the beam size is fixed to 10 and return generated sequences with scores. Scores are in the order of log-likelihood (negative), and the exponentiated scores are in the order of probability.

**Aggregator** We aggregate generations by taking the sum of generation probabilities. The model output with generation probabilities ($P_{LM}$) for each of the $K$ augmented prompts ($p_i$) will be fed into the aggregator to choose one final prediction. The aggregator recalculates the generation score ($s$) by taking the sum of the generation probabilities of identical generations (Eq.1). The final prediction of an object $y$ for the fact with subject $x$ and relation $r$ is the one with the highest score (Eq.2).

$$s(y'|x, r) = \sum_{i=1}^{K} P_{LM}(y'|p_i) \qquad (1)$$

$$y = \mathrm{argmax}(s(\cdot|x, r))_{y'} \qquad (2)$$

**Evaluation Metric** We quantify the effect of TTA as the relative change in exact match accuracy compared to not using TTA:

$$\text{relative effect} = \frac{(\#\text{correct w/ TTA}) + 1}{(\#\text{correct w/o TTA}) + 1} \qquad (3)$$

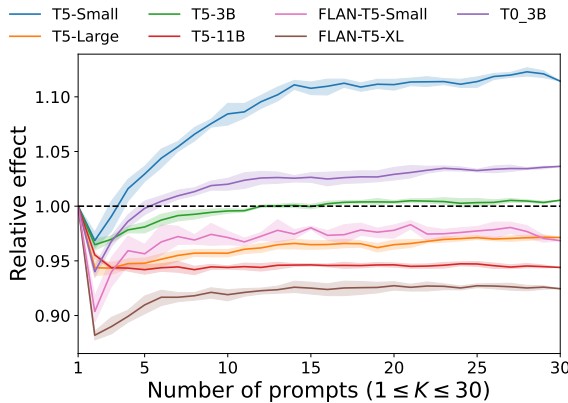

Figure 2: The relation between the number of prompts and the average relative effect (Eq.3) of TTA. A relative effect of 1.0 means no change in accuracy between with and without TTA.

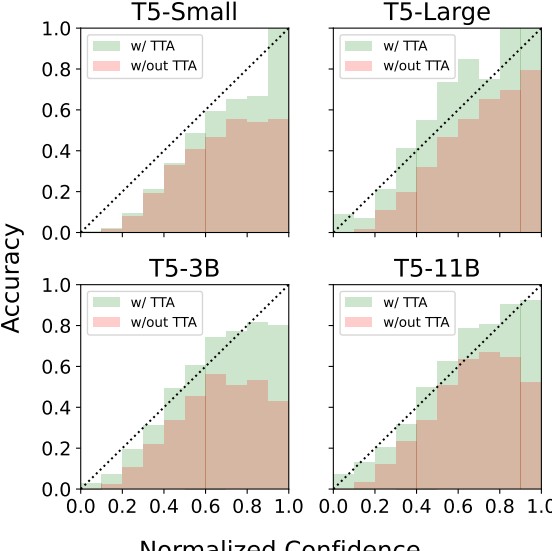

Figure 3: Comparison of model calibration with TTA and without TTA. TTA improves the model calibration of all four models in the comparison. Dotted lines represent ideal calibration, i.e., model confidence is equal to prediction accuracy.

To prevent division by zero, one is added to the numerator and the denominator. The metric judges correct only if the output is a perfect match. However, we observed cases where FLAN models are indifferent to capitalizations, e.g., the model generated "africa" instead of "Africa. For this reason, we exceptionally adopt case-insensitive match accuracy for the FLAN models.

## 3 Results

### 3.1 Positive Effects

**Accuracy**   The top half of Table 1 shows an example of TTA increasing the accuracy on the T5-11B model. The model gave an incorrect answer with the original prompt but produced the correct answer with TTA.

Here, we assess the impact of the number of prompts ($K$) on TTA performance. As described in §2, the augmenter produces 29 paraphrases from an original prompt. For each subset of $K$ prompts containing the original prompt and $K-1 (1 \leq K \leq 30)$ randomly sampled paraphrases, we measure the change in model accuracy in terms of relative effect (Eq.3).

We ran five iterations of random sampling and report the result in Figure 2. The sudden drop from $K = 1$[3] to $K = 2$ shows that models are highly sensitive to prompt variations. The relative effect appears to stabilize in the range of $20 \leq K \leq 30$ prompts, suggesting that aggregation of multiple prompts via TTA mitigates model sensitivity to variation of individual prompts. A comparison of

the result between $K = 1$ and $K = 30$ shows that TTA leads to better performance for three models, namely T5-Small, T5-3B, and T0_3B.

**Calibration**   In addition to increased accuracy, a second potential benefit of TTA is to improve model calibration by reducing overconfidence in incorrect answers. In this section, we investigate the effect of TTA on model calibration.

In our setup, the aggregator re-ranks generations by calculating the sum of the generation probability of identical generations for each fact instance. Because the original generation scores cannot be used as confidence after aggregation, we define the confidence of the aggregator as the ratio of the score to the final output and the sum of calculated scores for all candidate generations from all prompts for each fact instance.

$$\text{confidence} = \frac{\text{score}_{\text{final output}}}{\sum_{\text{candidates}} \text{score}} \qquad (4)$$

To enable a comparison of the relationship between the defined confidence and accuracy with and without TTA, we normalize scores to a maximum of 1. We then calculate the accuracy of model prediction with confidence between $0.1i$ and $0.1(i + 1)$ for each $0 \leq i < 10 \ (i \in \mathbb{N})$.

Figure 3 shows the relation between the calculated confidence and accuracy before and after ap-

---

[3] $K = 1$ setting indicates the baseline setting without TTA.

| Type | Prompt | Generation |
|------|--------|------------|
| Original | What continent is Para District located on? | Africa |
| WordNet | What continent is Para District based on? | North America |
| bt-fr | What continent is the Para District located on? | South America |
| bt-ru | What continent is Pará County on? | South America |
| bt-de | On which continent is the Para District located? | South America |
| Original | Where is Hans-Georg Gadamer buried? | Heidelberg |
| GLoVe | Accordingly is Hans-Georg Gadamer buried? | in Bonn |
| WordNet | Where is Hans-Georg Gadamer inhumed? | Erlangen |
| bt-fr | Where's Hans-Georg Gadamer buried. | Erlangen, Germany |
| bt-ru | Where's Hans-George Gadmer buried? | Wiesbaden, Baden-Württemberg |
| bt-de | Where's Hans-Georg Gadamer buried? | Erlangen, Germany |
| bt-es | Where is Hans-Georg Gadamer buried? | Heidelberg |
| bt-ja | Where are the goodly places? where is the plac... | Mount of Olives |
| Stopword-filtering | Where Hans-Georg Gadamer buried? | in Marburg |

Table 1: Examples of TTA improving and degrading performance. While Para District is located in " South America ", the model gives a wrong answer under the original prompt. Applying TTA yields the correct answer (top half). The bottom half of the table shows an example of TTA degrading performance. The correct answer is " Heidelberg ", but TTA yields " Erlangen, Germany ". The *Type* column specifies the augmentation type, with "GLoVe" and "WordNet" referring to synonym replacement and the "bt-" prefix indicating back-translation. Further examples are given in Appendix Table 3.

plying TTA. Without TTA, models tend to exhibit overconfidence by generating high-confidence answers with relatively low accuracy. This effect is especially pronounced in larger LMs, i.e., T5-3b and T5-11B. In contrast, after aggregation of multiple answers with TTA, confidence much better reflects accuracy since the accuracy of high-confidence predictions improved. These results show that TTA can improve model calibration.

## 3.2 Negative Effects

**Accuracy** Accuracy declines when the original prompt elicits the correct answer, but the TTA results in an incorrect answer. The bottom half of Table 1 shows an example of this. The 30 prompts yielded 18 unique model answers, among which seven prompts yielded the wrong answer "Erlangen, Germany", while only four prompted the correct answer "Heidelberg" (Table 1 shows only a subset of these answers). Overall, TTA led to performance degradation with the T5-Large, T5-11B, and the FLAN-T5 models (Figure 2).

**Error Analysis** Manual inspection suggests that the negative effects of TTA are mainly due to the low quality of the augmented prompts. Ideally, paraphrases should be grammatical, meaning-preserving, and exhibit variety in vocabulary choice and sentence structure. However, many augmented

prompts, such as those shown in the bottom half of Table 1, do not meet these criteria. For example, not all augmented prompts preserve the meaning of the original prompt.

To better understand the impact of the quality of automatically augmented prompts, we conducted additional evaluations. The first is to remove extremely poor paraphrases. Inspection of TTA errors revealed that one of the relations, namely "follows", was particularly error-prone in the augmentation step, likely due to the large variety of different entity and event types[4] for which this relation is used in Wikidata. We thus analyze the impact of TTA after removing all instances of the "follows" relation from the dataset. The second evaluation aims to improve the quality of paraphrases by using a large language model, namely GPT-3 text-davinci-003 (Brown et al., 2020), assuming that this larger LM produces better paraphrases than the simple augmentation methods used in our original setup. Figure 4 shows how GPT-3-based augmentation changed the effect of TTA on the T5 models. With the exception of T5-11B, augmentations produced by GPT-3 show a positive effect for all models.

---

[4]Arguments of the "follows" relation include: numbers (2 follows 1), months (February follows January) and events (the 2024 Paris Olympics follows the 2020 Tokyo Olympics).

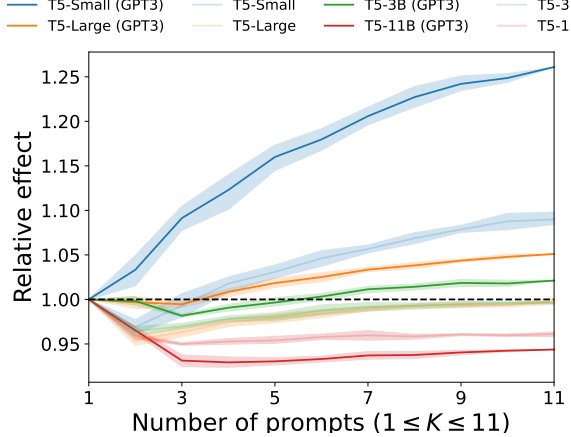

Figure 4: The relation between the number of prompts and the average relative effect (Eq.3) of TTA when removed "follows" relation and paraphrased using GPT-3.

## 4 Conclusions

We applied the idea of test-time augmentation (TTA) to language models, motivated by the observation that models are not robust to prompt variations. Specifically, we examined the effect of TTA on model accuracy and calibration in a factual probing setting.

Out of the seven models we investigated, TTA had a positive effect on the accuracy of smaller models, namely T5-Small, T5-3B, T0_3B. When controlling for the quality of augmentations, TTA also improved the accuracy of one more model (T5-Large). On other models, TTA had a negative effect in terms of accuracy. In terms of model calibration, we observed a positive effect since TTA reduced the number of high-confidence incorrect model answers.

The main remaining question is why the effect of TTA is inconsistent. We hypothesized that the inconsistent effect of TTA is due to the poor quality of automatically augmented prompts, and our analysis showed that the high quality of paraphrases is one of the important conditions for TTA to be effective. A related question we left to future work is how to enable TTA for relations that are difficult to paraphrase automatically, such as the "follows" relation (See Section 3.2).

While the ultimate goal is arguably to make language models robust against paraphrases without extensions like TTA, the results presented in this work show the potential benefit of TTA, especially for smaller LMs.

## Limitations

The TTA setup we present in this paper can be applied to short-answer generation tasks such as factual probing or classification tasks. A different setup would be necessary to apply TTA to tasks involving more complex model outputs, such as summarization or long-form question answering, since the aggregation step in our setup is not suited for such outputs.

In factual probing, TTA showed improvement in overall accuracy on three out of seven models. This means that whether or not TTA would improve the accuracy is unsure beforehand in practical uses. Deeper analyses of what improves/degrades the performance are needed to judge whether to use TTA or not.

## Ethics

Our experiments involved pre-trained language models and Wikidata, both of which are characterized by various forms of social biases. Consequently, our experiments and conclusions inherit and possibly amplify these biases.

## Acknowledgements

This work was supported by JST CREST Grant Number JPMJCR20D2, Japan, and JSPS KAKENHI Grant Numbers JP21K17814, JP21K21343, and JP22H00524. We would like to thank the members of TohokuNLP for their frequent participation in discussions during the course of this research.

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

# A Dataset

| Property ID | Property Name |
| --- | --- |
| P17 | Country |
| P19 | Place of birth |
| P20 | Place of death |
| P27 | Country of citizenship |
| P30 | Continent |
| P36 | Capital |
| P37 | Official language |
| P50 | Author |
| P69 | Educated at |
| P103 | Native language |
| P119 | Place of burial |
| P131 | Located in the administrative territorial entity |
| P140 | Religion or worldview |
| P155 | Follows |
| P156 | Followed by |
| P159 | Headquarters location |
| P407 | Language of work or name |
| P495 | Country of origin |
| P641 | Sport |
| P740 | Location of information |
| P937 | Work location |
| P1365 | Replaces |
| P1366 | Replaced by |
| P1376 | Capital of |
| P1412 | Languages spoken, written, or signed |

Table 2: Wikidata property IDs and English labels used for constructing our dataset.

We collected 500 relational facts for each of the 25 relations selected from WikiData. Table 2 shows the corresponding Wikidata property ID and English property names.

## B  Augmentation Methods

**Synonym Replacement**  We use the Python library "TextAttack" to replace words with their synonyms. Specifically, for synonym replacement using WordNet, we use the WordNetAugmenter, and for synonym replacement using GloVe embeddings, we use the WordSwapEmbedding class.

**Back-translation**  We first translate the original prompt to eight candidates in the target language. Each candidate is then translated back into eight candidates in the source language, resulting in a total of 64 back-translated prompt candidates. We adopt the round-trip probability as the score of the back-translated prompt candidates and select four candidates using the aggregation method mentioned in Section 2. For translations, we used OPUS-MT models (Tiedemann and Thottingal, 2020) The OPUS-MT models occupy roughly the same memory size as the T5-Small model.

**Stopwords-filtering**  This method removes stopwords and diacritics from the original prompt using the Python library "Texthero".

## C   Count-based Aggregation of Model Answers

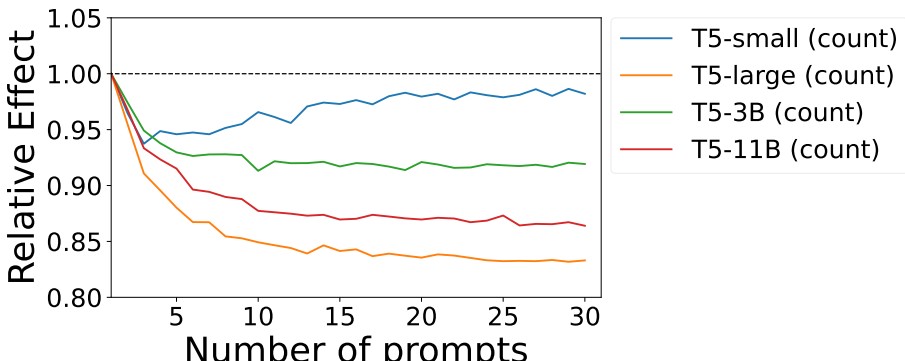

Figure 5: The relation between the number of prompts and the average relative effect of TTA when aggregated using a count-based method. A relative effect of 1.0 means no change in accuracy between with and without TTA.

Counting the number of appearances in the generations is one method of aggregation. We did not use count-based aggregation in our main experiments because the possibility of having multiple generations with the same counts is high. The phenomenon is predicted to occur more often when we increase the number of sequences the model outputs. In addition, this method cannot take confidence into account as all generations by beam-search are equally weighted.

The result of using count-based aggregation is shown in figure 5. TTA degrades model performance with count-based aggregation.

# D  Additional output examples

Table 3 gives a complete example of all augmented prompts and corresponding model outputs. (A subset of these is also shown in Table 1 in the main part of the paper).

| Type | Prompt | Generation |
|---|---|---|
| Original | Where is Hans-Georg Gadamer buried? | Heidelberg |
| GLoVe | Accordingly is Hans-Georg Gadamer buried? | in Bonn |
| GLoVe | Consequently is Hans-Georg Gadamer buried? | in Bonn |
| GLoVe | Where poses Hans-Georg Gadamer buried? | Erlangen |
| GLoVe | Where represents Hans-Georg Gadamer buried? | Erlangen, Germany |
| WordNet | Where is Hans-Georg Gadamer immerse? | German philosophical neo-Kantia |
| WordNet | Where is Hans-Georg Gadamer inhume? | Erlangen, Germany |
| WordNet | Where is Hans-Georg Gadamer inhumed? | Erlangen |
| WordNet | Where is Hans-Georg Gadamer swallow? | the throat |
| bt-fr | Where is Hans-Georg Gadamer buried? | Heidelberg |
| bt-fr | Where's Hans-Georg Gadamer buried? | Erlangen, Germany |
| bt-fr | Where's Hans-Georg Gadamer buried. | Erlangen, Germany |
| bt-fr | Where's Hans-Georg Gadamer buried in the house? | in the garden |
| bt-ru | Where is Hans-George Gadmer buried? | Basel |
| bt-ru | Where is Hans-George Gadamer buried? | Erlangen, Germany |
| bt-ru | Where's Hans-George Gadmer buried? | Wiesbaden, Baden-Württemberg |
| bt-ru | Where is Hans-Gorg Gadmer buried? | Wiesbaden, Baden-Württemberg |
| bt-de | Where's Hans-Georg Gadamer buried? | Erlangen, Germany |
| bt-de | Where is Hans-Georg Gadamer buried? | Heidelberg |
| bt-de | Where was Hans-Georg Gadamer buried? | in the Munich Waldfriedhof |
| bt-de | Where will Hans-Georg Gadamer be buried? | in the Cathedral of Speyer |
| bt-es | Where is Hans-Georg Gadamer buried? | Heidelberg |
| bt-es | Where's Hans-Georg Gadamer buried? | Erlangen, Germany |
| bt-es | Where is Hans-Georg Qadamer buried? | In a Munich cemetery |
| bt-es | Where is Hans-Georg Gadhamer buried? | Innsbruck |
| bt-ja | Where are the goodly places? where is the plac... | Mount of Olives |
| bt-ja | Where are the goodly places? Where is the plac... | Bethel |
| bt-ja | Where are the goodly places? where are the pla... | the mountain of God |
| bt-ja | Where are the goodly places? where is the plac... | the place of his fathers |
| Stopword-filtering | Where Hans-Georg Gadamer buried? | in Marburg |

Table 3: An example of a case in which TTA degrades the accuracy of LM answers. The correct answer to the question "Where is Hans-Georg Gadamer buried?" is " Heidelberg ", but the aggregator returned " Erlangen, Germany ".

# E GenBench Evaluation Card

To situate our work in the broader context of efforts to understand and improve the generalization of machine learning models for natural language processing, Table 4 provides a GenBench evaluation card (Hupkes et al., 2022).

| Motivation | | | | | |
|---|---|---|---|---|---|
| *Practical* ☐ | | *Cognitive* | | *Intrinsic* | *Fairness* |
| **Generalisation type** | | | | | |
| *Compositional* | *Structural* | *Cross Task* | *Cross Language* | *Cross Domain* | *Robustness* ☐ |
| **Shift type** | | | | | |
| *Covariate* ☐ | | *Label* | | *Full* | *Assumed* |
| **Shift source** | | | | | |
| *Naturally occuring* | | *Partitioned natural* | *Generated shift* ☐ | | *Fully generated* |
| **Shift locus** | | | | | |
| *Train–test* | | *Finetune train–test* | *Pretrain–train* | | *Pretrain–test* ☐ |

Table 4: GenBench Evaluation Card