# OpenReview forum: "Test-time Augmentation for Factual Probing"
_EMNLP/2023/Conference — EMNLP 2023 Findings_

### Official Review · Reviewer_jtoE · 2023-07-20

**Soundness:** 3

**Excitement:**

2: Mediocre: This paper makes marginal contributions (vs non-contemporaneous work), so I would rather not see it in the conference.

**Paper Topic And Main Contributions:**

This work proposes a prompt augmentation method TTA without training instances. The augmentation includes synonym substitution, back-translation and stop word filtering. They collected a relational dataset from Wikidata and evaluated several pretrained language models of different sizes, and observe that TTA introduces more positive impacts for pretrained models of smaller sizes. And TTA in general improves calibration of the model. Lastly, they find that prompt rephrasing from GPT-3 also has benefits to most model variants.

**Questions For The Authors:**

- How many relations are there in the dataset evaluated (what are them), and how many instances does each relation has?
- Does the x-axis of Figure 2 starts at 0 or 1?
- What are the order of prompts in the x-axis of Figure 2, 4 and 5? Since there are prompts from different augmentation approaches, it is hard to identify the performance of each category. Especially when there is a performance drop immediately after the start.
- What are the candidates generated in line 170, the generation from all prompts, or the generations from the same beam of individual prompts?

**Reasons To Accept:**

The test time prompt augmentation is an interesting problem to improve the effectiveness prompt without training. The proposed method is simple to generalize.

**Reasons To Reject:**

- The back-translation based prompt rephrasing is also used in [1].
- The introduction of the configuration and results lack important details to fully understand the procedure and performance.
- The pretrained model of larger sizes are observed to demonstrate stronger ability of language comprehension and therefore more robust to the variance in the prompt templates, which would undermines of importance of ensemble multiple prompts generated from linguistic perturbations, The results also show that models of larger sizes receive lighter improvement corresponds to the smaller counterparts.


[1] Jiang et al. How can How can we know what language models know?

**Reproducibility:**

N/A: Doesn't apply, since the paper does not include empirical results.

**Reviewer Confidence:**

4: Quite sure. I tried to check the important points carefully. It's unlikely, though conceivable, that I missed something that should affect my ratings.

---

> ### Author Rebuttal · Authors · 2023-08-29
>
> We are pleased that the reviewer finds our test time prompt augmentation approach both interesting and appreciable in its simplicity. We hope our following answers address the raised questions and concerns satisfactorily (Reasons to Reject 2).
>
>
> ## 1. About Jiang et al. (2020)’s work (Reasons to Reject 1):
>
> We acknowledge that the back-translation method is well-established, having been used in the MT community for years [1][2]. Our contribution lies in the utilization of it to evaluate the test-time augmentation method for factual probing tasks.
> Unlike Jian et al. (2020), our approach does not rely on pre-selecting paraphrases using a pre-gathered dataset. Our TTA overcomes this constraint, offering spontaneous, relation-agnostic augmentation.
> We will improve the writing to clarify this in our final draft.
>
> [1] Ondrej Bojar and Ales Tamchyna. 2011. Improving translation model by monolingual data. In Workshop on Statistical Machine Translation (WMT), and Rico Sennrich, Barry Haddow, and Alexandra Birch. 2016.
> [2] Improving neural machine translation models with monolingual data. Conference of the Association for Computational Linguistics (ACL).)
>
>
> ## 2. About the impact of our approach to large models (Reasons to Reject 3):
>
> If we assume that larger LMs are generally "better", then we can also assume that they are more robust to meaning-preserving input variations, so it is expected that the impact of our method becomes smaller as model size increases. In this study, we empirically assess the effectiveness of the test-time augmentation method for factual probing tasks across various model sizes. Rather than solely pursuing state-of-the-art results with larger models, it is also important to improve smaller LMs (environmental impact, financial constraints, hardware constraints when running LMs on edge devices, etc) and this is exactly where our method works best.
>
> We would also like to remind that the approach is effective in calibrating model confidence to model accuracy as explained using Figure 3, even (or especially) on large models. For example, before implementing TTA, T5-11B model had low accuracy when the it generated the answers with high confidence(right part of the orange line in the lower right graph in Figure 3). This means a considerable number of answers were over-confident and incorrect. After TTA, the relationship between the model accuracy and the confidence aligns better to the ideal line (the diagonal dotted line in the figure). This indicates that TTA reduced over-confident and incorrect answers.
>
> ## 3. About the dataset we constructed (Question 1):
> Thank you for noticing this oversight. We gave only the total number of relation instances in the submission and indeed did not include counts for each relation. We collected 500 instances from each of 25 relations we selected, giving the total of X * Y = 12,500 relational facts we stated on line 82 in the submission. All instances are collected from WikiData, and the selected relations (properties) are the following. The relations were selected under the condition that we can collect 500 instances with low distribution bias. We will also add the information to the paper when it is accepted.
>
> |Property ID | Property Name |
> | - | - |
> |P17  | Country |
> |P19  | Place of birth |
> |P20  | Place of death |
> |P27  | Country of citizenship |
> |P30  | Continent  |
> |P36  | Capital |
> |P37  | Official language |
> |P50  | Author |
> |P69  | Educated at |
> |P103  | Native language |
> |P119  | Place of burial |
> |P131  | Located in the administrative territorial entity  |
> |P140  | Religion or worldview |
> |P155  | Follows |
> |P156  | Followed by |
> |P159  | Headquarters location |
> |P407  | Language of work or name |
> |P495  | Country of origin |
> |P641  | Sport |
> |P740  | Location of information |
> |P937  | Work location |
> |P1365  | Replaces |
> |P1366  | Replaced by |
> |P1376  | Capital of |
> |P1412  | Languages spoken, written, or signed |
>
>
> ## 4. About the starting value of x-axis in Figure 2 (Question 2):
> Thank you for noticing this oversight. The x-axis in Figures 2, 4, and 5 all starts with 1. The points where the x (number of prompts) is 1 represent the accuracy of the model with no prompt augmentation. We will revise the Figures in our camera ready.
>
>
> ## 5. About the order of prompts in Figures 2, 4, and 5 (Question 3):
> If the “order” mentioned refers to the scale of the x-axis, it is in a linear scale. 1 means that one single prompt is inputted into the model, and 10 means that 10 prompts, including one original prompt, are fed to the model.
> If the “order” refers to “how we chose the 5, for example, prompts”, it was randomly sampled from the pool of augmented prompts. In the setting of Figure 2, there are one original prompt and 29 augmented prompts for each instance of the dataset. The plot where the x value is 5 means one original prompt is ensembled with 4 other randomly sampled augmented prompts.
>
> ## 6. About “candidates generated” in line 170 (Question 4):
> It refers to “all the generation from all prompts”. Since we fixed the beam size to 10, if the number of prompts was $P$, there would be $10\times P$ candidates.

---

### Official Review · Reviewer_HiLF · 2023-08-03

**Soundness:** 4
**Typos Grammar Style And Presentation Improvements:** 1）Line 061 "Nevertheless, TTA is hard…

**Excitement:**

4: Strong: This paper deepens the understanding of some phenomenon or lowers the barriers to an existing research direction.

**Paper Topic And Main Contributions:**

This paper aims to address the problem in the factual probing  task, which is that small changes to the prompt can lead to large changes in model output. The authors propose to use test-time augmentation (TTA) as a relation-agnostic method for reducing sensitivity to prompt variations by automatically augmenting and ensembling prompts at test time.

**Reasons To Accept:**

Experiments show model confidence better reflects prediction accuracy by improving model calibration. The results presented in this work show the potential benefit of TTA, especially for smaller LMs.

**Reasons To Reject:**

Adaptability to large models limits the potential value of this technique.

**Reproducibility:**

3: Could reproduce the results with some difficulty. The settings of parameters are underspecified or subjectively determined; the training/evaluation data are not widely available.

**Reviewer Confidence:**

3: Pretty sure, but there's a chance I missed something. Although I have a good feel for this area in general, I did not carefully check the paper's details, e.g., the math, experimental design, or novelty.

---

> ### Author Rebuttal · Authors · 2023-08-29
>
> We are pleased that the reviewer acknowledges that the contribution of our work is in improving model reliability by applying TTA and improving model accuracy or calibration of model confidence to accuracy. We hope our following answers address the raised questions and concerns satisfactorily.
>
> ## 1. Impact of our approach to large models (Reasons to Reject 1):
> If we assume that larger LMs are generally "better", then we can also assume that they are more robust to meaning-preserving input variations, so it is expected that the impact of our method becomes smaller as model size increases. In this study, we empirically assess the effectiveness of the test-time augmentation method for factual probing tasks across various model sizes. Rather than solely pursuing state-of-the-art results with larger models, it is also important to improve smaller LMs (environmental impact, financial constraints, hardware constraints when running LMs on edge devices, etc) and this is exactly where our method works best.
>
> We would also like to remind the reviewer that the approach is effective in calibrating model confidence to model accuracy as explained using Figure 3, even (or especially) on large models. For example, before implementing TTA, T5-11B model had low accuracy when it generated the answers with high confidence(right part of the orange line in the lower right graph in Figure 3). This means a considerable number of confident answers were incorrect (= over-confident). After TTA, the relationship between the model accuracy and the confidence aligns better to the ideal line (the diagonal dotted line in the figure). This indicates that TTA reduced over-confident and incorrect answers.
>
> ## 2. Why TTA is hardly used in NLP tasks (Question 1):
>
> We will add an explanation to the paper in the camera ready. Our explanation on why TTA is hardly used in NLP tasks would be the following.
> TTA is easy to use with visual data since there is basically an infinite amount of augmentations that are almost guaranteed to preserve the input features in question. For example, a flipped image of a cat is still an image of a cat, the same for rotation, color variations, adding noise, blurring, etc. In contrast, automatic augmentation of textual data is much more difficult.
>
> ## 3. About $K$ in Equation 1 (Question 2):
> We appreciate the reviewer for noticing the oversight. The $K$ stands for the number of prompts we feed to the model for each instance. For example, if we obtain 3 paraphrases from an original prompt, $K$ equals 4, which is the sum of 3, the number of paraphrases, and 1, the number of original prompt.
>
> ## 4. About the reason we adopt case-insensitive match accuracy on FLAN models (Question 3):
> We will add an explanation to the paper when it is accepted.
> The reason why we adopt case-insensitive match accuracy in FLAN models is because we found numerous outputs that ignored capitalizations. For example, there were cases where the model generated “africa” instead of “Africa”, which was not observed in other models we tested.

---

### Official Review · Reviewer_nPms · 2023-08-05

**Soundness:** 3

**Excitement:**

3: Ambivalent: It has merits (e.g., it reports state-of-the-art results, the idea is nice), but there are key weaknesses (e.g., it describes incremental work), and it can significantly benefit from another round of revision. However, I won't object to accepting it if my co-reviewers champion it.

**Paper Topic And Main Contributions:**

The factual probing results could change as the prompts change, making the probing results unstable. This paper mitigates this problem by using more prompts to query each fact. The additional prompts are obtained by synonym replacement. The proposed methods can somewhat improve the factual probing accuracy of some datasets.

**Reasons To Accept:**

This paper works on an important problem in factual probing, i.e., models could give quite different results for the same sample due to the small changes in prompts.

**Reasons To Reject:**

The proposed augmentation method is not very novel. A similar approach has been proposed in "Self-Consistency Improves Chain of Thought Reasoning in Language Models."

The experiments show the "impact of the number of prompts" and the "relation between the calculated confidence and accuracy". However, in the abstract, the motivation is alleviating the problem that "small changes to the prompt can lead to large changes in model output." I do not understand how the proposed method helps to solve the above "small change ..." problem.

**Reproducibility:**

4: Could mostly reproduce the results, but there may be some variation because of sample variance or minor variations in their interpretation of the protocol or method.

**Reviewer Confidence:**

2: Willing to defend my evaluation, but it is fairly likely that I missed some details, didn't understand some central points, or can't be sure about the novelty of the work.

---

> ### Author Rebuttal · Authors · 2023-08-29
>
> We are pleased that the reviewer recognizes the significance of the problem addressed in our paper. We hope our subsequent responses adequately address the questions and concerns raised.
>
>
> ## 1. Wang et al. (2022): Self-Consistency Improves Chain of Thought Reasoning in Language Models (Reasons to Reject 1):
>
> Thank you for your reference. While Wang et al.'s motivation to generate reliable outputs aligns with ours, their approach differs. Wang et al. suggest generating multiple outputs from a single prompt and then taking the majority votes. In contrast, our method involves generating outputs from multiple prompts and then taking the majority votes.
> Our contribution lies in empirically demonstrating the effectiveness (or lack thereof) of the test time augmentation method for the factual probing task. Instead of merely chasing the state-of-the-art, we believe that it is crucial to explore similar yet distinct approaches and share the results with the community.
>
> ## 2. Small changes to the prompt can lead to large changes in model output (Reasons to Reject 2):
>
> We are emphasizing that a **single** alteration of a prompt can influence a model's outputs. Through test time augmentation, we can produce multiple outputs from **several** slightly modified prompts. By taking the majority/average, we can mitigate the impact of one minor alteration.
> The sentence in the abstract is a little unclear. We will rectify this in the final draft.

---

### Meta-Review · Area_Chair_cekW · 2023-09-19

**Recommendation:** 3

**Metareview:**

Paper Topic And Main Contributions:
* This paper aims to address the problem in the factual probing task, which is that small changes to the prompt can lead to large changes in model output.
* The authors propose to use test-time augmentation (TTA) as a relation-agnostic method for reducing sensitivity to prompt variations by automatically augmenting and ensembling prompts at test time.
* The augmentations include synonym substitution, back-translation and stop word filtering.

Reasons to accept:
* The problem of answer variability depending on the prompt is important.
* The test time prompt augmentation is an interesting solution to improve the effectiveness prompt without training. The proposed method is easy to generalize.

Reasons to reject:
* The proposed augmentation method is not very novel. A similar approach has been proposed in "Self-Consistency Improves Chain of Thought Reasoning in Language Models." The authors explain in the rebuttal: Wang et al. suggest generating multiple outputs from a *single* prompt and then taking the majority vote. In contrast, the proposed method involves generating outputs from *multiple* prompts and then taking the majority vote.
* Diminishing improvements for larger models. The authors explain in the rebuttal that smaller LMs are important (environmental impact, financial constraints, hardware constraints when running LMs on edge devices, etc).

---

### Decision · Program_Chairs · 2023-10-07

**Decision:**

Accept-Findings

**Comment:**

Paper Topic And Main Contributions:
* This paper aims to address the problem in the factual probing task, which is that small changes to the prompt can lead to large changes in model output.
* The authors propose to use test-time augmentation (TTA) as a relation-agnostic method for reducing sensitivity to prompt variations by automatically augmenting and ensembling prompts at test time.
* The augmentations include synonym substitution, back-translation and stop word filtering.

Reasons to accept:
* The problem of answer variability depending on the prompt is important.
* The test time prompt augmentation is an interesting solution to improve the effectiveness prompt without training. The proposed method is easy to generalize.

Reasons to reject:
* The proposed augmentation method is not very novel. A similar approach has been proposed in "Self-Consistency Improves Chain of Thought Reasoning in Language Models." The authors explain in the rebuttal: Wang et al. suggest generating multiple outputs from a *single* prompt and then taking the majority vote. In contrast, the proposed method involves generating outputs from *multiple* prompts and then taking the majority vote.
* Diminishing improvements for larger models. The authors explain in the rebuttal that smaller LMs are important (environmental impact, financial constraints, hardware constraints when running LMs on edge devices, etc).